# Accounting for Area Sources in Air Pollution Models

**DOI:** 10.3390/ijerph20126110

**Published:** 2023-06-12

**Authors:** Akula Venkatram, Ranga Rajan Thiruvenkatachari

**Affiliations:** Department of Mechanical Engineering, University of California, Riverside, CA 92521, USA

**Keywords:** area sources, line sources, dispersion model, inverse modeling, methane emissions, manure lagoons, dairy emissions, emission uncertainty

## Abstract

Area sources are important components of comprehensive air pollution models. The literature describes several approaches to modeling dispersion from such sources, but there is little consensus on an approach that can be applied to arbitrarily shaped area sources and is numerically efficient at the same time. This paper brings together ideas from previous work to propose an approach that meets these requirements. It is based on representing an area source as a set of line sources perpendicular to the wind direction; the number of line sources is determined by the specified precision of the concentration computed at a receptor impacted by the area source. Although AERMOD and the OML model incorporate versions of this approach, the open literature lacks an adequate description. This paper fills this important gap and also provides examples of its application. We show that different shaped area sources with the same emissions and emission density yield significantly different downwind concentration patterns. We then demonstrate the utility of the method through inverse modeling to estimate methane emissions from manure lagoons located in a dairy.

## 1. Introduction

A variety of sources of pollutant emissions can be modeled as area sources. These include manure lagoons, landfills, wastewater treatment ponds, and highways. A group of point sources can also be treated as an area source. Several approaches have been used to model dispersion of emissions from such sources. Hanna [1] modeled an urban area as a set of area sources. Dispersion of emissions from each source is modeled by assuming that emissions are uniformly mixed over a grid square. This yields reasonable results if the receptor is located within an emission grid so that the concentration is dominated by emissions within a grid square. Another approach assumes that concentrations downwind of an area source can be estimated by assuming that the area source is a point source with initial plume spread determined by the horizontal dimensions of the area source. The validity of this approximation has not been tested when the receptor is close to the source. The open literature presents a small number of papers [2,3,4,5] that treat the area source more accurately. In the approach described in these papers, the area source is divided into narrow strips, each of which is then modeled as a finite line source in computing their contributions to the concentration at a receptor. This approach is restricted to rectangular area sources. The authors of [6] provide a brief description of the line source approach adapted for area sources that need not be rectangular. This method is implemented in the AERMOD [7] modeling system. The Danish OML model [8] uses a similar method in modeling area sources.

The peer-reviewed literature lacks a complete description of an approach to treat an area source with an arbitrary shape. This paper fills this gap by first providing details of the formulation of the area source problem. While the formulation builds upon previous approaches, it incorporates some improvements. We then describe the numerical methods used to implement the formulation.

We demonstrate the application of the area source model by inferring methane emissions from manure lagoons in a dairy farm. Methane has an increasingly important role in causing climate change, and its emissions are rising more quickly than those of CO_2_ [9]. Animal agriculture is the source of ~35% of anthropogenic methane emissions globally, and these emissions are increasing along with the number of animals [9]. Methane emissions from animal agriculture derive primarily from enteric fermentation in cattle and from manure management, particularly when waste is treated or stored in anaerobic lagoons. In a previous paper [10], we developed and applied an area source dispersion model to estimate methane emissions from manure lagoons located in a southern California dairy. Here, we provide a summary of the results to illustrate a real-world application of the area source dispersion model.

## 2. Representation of Area Source

Figure 1 depicts dispersion of emissions from an area source located at a height $z_{s},$ and emitting a pollutant at a total rate of $Q\;\left(mass/time\right)$. Our objective is to compute the steady state concentration, $C\;\left(mass/volume\right)$, of the pollutant at the receptor located at $\left(x_{r},y_{r},\;z_{r}\right)$. We assume that both $z_{s}$ and $z_{r}$ are well within the surface boundary layer, which can be described with profiles of horizontal velocity, $U\left(z\right)$, and eddy diffusivity, $K\left(z\right)$. A later section provides more details of these profiles.

Figure 2 shows the top view of the area source represented as a concave or a convex polygon with an arbitrary number of sides. We use a coordinate system in which the *x*-axis is along the near surface wind vector. The contribution of the area source to the concentration at a receptor at $\left(x_{r},y_{r},z_{r}\right)$ is computed by representing the area by a set of equally spaced finite line sources that span the along-wind width of the area source. The number of line sources is determined by the required precision of the concentration computed at the receptor of interest.

The finite line source is the basic building block of the area source model. Each line source spans the area source and is placed perpendicular to the near surface wind speed (Figure 2). To see why the line sources must be perpendicular to the wind direction, let us first assume that the area source is represented by a fixed set of line sources that are not perpendicular to the wind direction. Figure 3 shows one of the line sources inclined at an angle, θ, to the mean wind.

The beginning and end points of the line source are at $\left(x_{b},y_{b},z_{s}\right)$ and $\left(x_{e},y_{e}.z_{s}\right)$. The concentration at a receptor is the integral of the contributions of the elemental sources, along the line source, each of which has a different downwind distance relative to the receptor, $\left(x_{r}-x\right)$. The source strength of an elemental source at $\left(x,y,z_{s}\right)$ is qds, where q is the emission rate per unit length of the source, and ds is the length of the element, ds=dy/cosθ. Then, the concentration at $\left(x_{r},y_{r},z_{r}\right)$ associated with the line source is
(1)$$C\left(x_{r},y_{r},z_{r}\right)=q\int_{y_{b}}^{y_{e}}F_{z}\left(x_{r}-x,z_{s},z_{r}\right)\frac{1}{\sqrt{2\pi }}\exp\left(-\frac{{\left(y-y_{r}\right)}^{2}}{2\sigma_{y}^{2}\left(x_{r}-x\right)}\right)\frac{dy}{\cos\theta },$$
where the horizontal distribution is taken to be Gaussian. In Equation (1), $F_{z}$ is the vertical distribution of the concentration for unit source strength, in which $z_{s}$ is the height of source, and $z_{r}\;$ is the receptor height. $F_{z}$ is a function of meteorology as well as source and receptor coordinates. If $F_{z}$ is described by a Gaussian distribution and the wind speed is constant with height or represented by an effective value, $U_{e}$, we have
(2)$$F_{z}\left(x_{r}-x,z_{s},z_{r}\right)=\frac{1}{\sqrt{2\pi }\sigma_{z}\left(x_{r}-x\right)U_{e}}\left[\exp\left(-\frac{{\left(z_{s}-z_{r}\right)}^{2}}{2\sigma_{z}^{2}\left(x_{r}-x\right)}\right)+\exp\left(-\frac{{\left(z_{s}+z_{r}\right)}^{2}}{2\sigma_{z}^{2}\left(x_{r}-x\right)}\right)\right].$$

In the model presented in this paper, we describe $F_{z}$ using the numerical solution of the advection-diffusion equation, which allows us to account explicitly for the variation of the wind speed with height rather than represent the wind speed by an effective value.

This integral has to be evaluated numerically because $\left(x_{r}-x\right)$ is a function of the y coordinate. Ref. [11] provides an approximate analytical expression that is accurate at angles ≤ $80^{{}^{\circ}}.$ However, it is an approximation that has errors at large θ and must account for some part of the line source being downwind of the receptor.

Because the directions of the line sources used to compute the area contribution can be chosen arbitrarily, we can avoid the problems associated with a finite θ, by drawing the lines perpendicular to the mean wind, so that θ=0, and $\left(x_{r}-x\right)$ does not vary with y. Then, Equation (1) can be integrated analytically to yield the expression
(3)$$\begin{array}{l} C\left(x_{r},y_{r},z_{r}\right)=qF_{z}\left(x_{r}-x_{l},z_{s},z_{r}\right)\left(erf\left(t_{2}\right)-erf\left(t_{1}\right)\right)/2\\ \mathrm{where}\\ t_{1}=\frac{y_{r}-y_{b}}{\sqrt{2}\sigma_{y}\left(x_{r}-x_{l}\right)},\;\mathrm{and}\;t_{2}=\frac{y_{r}-y_{e}}{\sqrt{2}\sigma_{y}\left(x_{r}-x_{l}\right)} \end{array}$$
where $x_{l}$ is the x coordinate of the line source that is perpendicular to the wind direction (Figure 2). The vertical distribution of the concentration, $F_{z}$, does not vary along y. One convenient approach to calculating this distribution is described next.

## 3. Vertical Distribution

The vertical distribution $F_{z}$ is commonly described with a Gaussian distribution (AERMOD [7] for example). We suggest the solution of the two-dimensional diffusion equation as an alternative because it offers several advantages that are discussed in this section. The solution provides an excellent description of the vertical distribution of tracer concentrations measured during the classic Prairie Grass experiment [12] when the wind speed and eddy diffusivity are formulated using the Monin–Obukhov similarity [13]. The solution, unlike the Gaussian distribution, describes the evolution of the distribution from nearly Gaussian during stable conditions to nearly exponential during unstable conditions. Note that the current formulations of plume spread in a Gaussian framework [14] draw upon this early work in the paper by [13].

The vertical distribution of concentrations is computed using the mass conservation equation expressed in terms of the crosswind integrated concentration, ${\bar{C}}^{y}$, which we denote by C here for convenience
(4)$$U\left(z\right)\frac{\partial C}{\partial x}=\frac{\partial }{\partial z}\left(K(z)\frac{\partial C}{\partial z}\right),$$
where $K\left(z\right)$ is the vertical eddy diffusivity, and $U\left(z\right)$ is the horizontal velocity. We take the source, *q (mass/*(*time.length*)) to be located at $\;z=z_{s}$ at x=0. The boundary conditions are
(5)$$\begin{array}{l} K\left(z\right)\frac{\partial C}{\partial z}=-v_{d}C\;\mathrm{at}\;z=z_{0}\\ \mathrm{and}\\ \frac{\partial C}{\partial z}=0\;\mathrm{at}\;z=H \end{array}$$
where $v_{d}$ is the deposition velocity, $z_{0}$ is the roughness length, and H is the top of the modeling domain.

The mass conservation Equation (4) models turbulent dispersion using the concept of eddy diffusivity, which can be justified only when the travel time from the source is much larger the relevant Lagrangian time scale that governs particle motion in the turbulent flow [15]. Its success in describing dispersion from surface releases provides a posteriori justification for its use.

The horizontal plume spread, $\sigma_{y}$, used in the expression for the contribution of a line source is based on the expression suggested in [16] and applied in [14] to describe horizontal spread of plumes released during the Prairie Grass field study:(6)$$\begin{array}{l} \frac{d\sigma_{y}}{dx}=\frac{\sigma_{v}}{U\left(\bar{z}\right)}\\ \mathrm{where}\\ \bar{z}\left(x\right)=\frac{\int_{0}^{H}zC\left(x,z\right)dz}{\int_{0}^{H}C\left(x,z\right)dz} \end{array}$$

The mean height, $\bar{z}$, is computed from the concentration profile, $C\left(z\right),$ obtained from the numerical solution of Equation (4).

## 4. Numerical Methods

The horizontal domain for the solution of Equation (4) is taken to be 1.1 times the maximum distance between the vertex of the area source and the receptors. The vertical domain is taken to be several times larger than the estimated vertical spread of the plume at the maximum source-receptor distance; the vertical spread is limited by the height of the mixed layer. The horizontal grid points are linearly spaced with fine enough spacing to allow the use of two-dimensional linear interpolation to compute concentrations at receptors that do not coincide with grid points. The vertical spacing is logarithmic to provide fine resolution close to the surface.

Equation (4) is solved numerically using an upwind difference scheme for the advection component on the left-hand side of the equation. The diffusion term on the right is represented with a three-point finite difference scheme. The line source is specified with a Gaussian concentration distribution centered at the source height with a nominally small vertical spread. The mass flux at a convenient distance, usually three grid points from the origin, is computed as a vertical integral of the product of the velocity and the concentration. This mass flux is then used to normalize the concentrations to yield that corresponding to unit emission rate.

The contribution of the area source to the concentration at a receptor is computed by summing over contributions from equally spaced line sources that span the maximum along-wind width of the area source. The contribution of a set of N line sources on an area source to the concentration at a receptor is written as
(7)$$C\left(x_{r},y_{r},z_{r}\right)=q\sum_{i=1}^{i=N}F_{z}\left(x_{r}-x_{l}^{i},z_{s},z_{r}\right)F_{y}\left(x_{r}-x_{l}^{i},y_{r}-y_{b}^{i},y_{r}-y_{e}^{i}\right),$$
where $x_{l}^{i}$ is the x-co-ordinate of the $i^{th}$ line source with a length $l^{i}$ and $F_{y}$ is the horizontal distribution of concentration (Equation (3)). The sum in Equation (7) is conducted in steps in which the number of line sources are inserted between the existing set of lines. The value of the sum in each step is combined with that calculated in the previous step by weighing each of them by the corresponding total length of the line sources. We illustrate the process by writing Equation (7) as
(8)$$C\left(x_{r},y_{r},z_{r}\right)=q\sum_{i=1}^{i=N}f^{i},$$
where $q=\frac{Q}{p},\;p=\sum_{i=1}^{i=N}l^{i}$, and $f^{i}$ is the contribution of the $i^{th}$ line source, with a length $l^{i}$, to the concentration at the receptor. We have combined $F_{z}$ and $F_{y}$ and dropped the dependence on $\left(x,\;y,z\right)$ for convenience. The emission rate per unit length, q, is expressed in terms of the total emission rate from the source, Q, and the sum, p, of the length of the line sources,$\;l^{i}$.

The updating of concentrations by inserting line sources between existing line sources is conducted using the following steps (Figure 4). First, the domain over which the line sources are placed is determined by identifying the maximum, $x_{max},$ and minimum, $x_{min}$, of the coordinates of the source, in the coordinate system in which the *x*-axis lies along the mean wind direction. In the first step, j=1, a line source is placed mid-point, $x_{mid}$, between $x_{max}$ and $x_{min}$, and the corresponding contribution is computed at the receptor. In step j=2, 2 line sources are placed, one halfway between $x_{min}\;$ and $x_{mid}$ and another halfway between $x_{mid}$ and $x_{max}$. In j=3, 4 line sources are placed each midway between the line sources placed in j=2. So, the relationship between the number of line sources, $n_{j}$ placed in step ‘j’ between $x_{min}\;$ and $x_{max}$, is $n_{j}=2^{j-1},\;$ and the total number of line sources, $N_{j}$, that are on the line joining $x_{min}$ and $x_{max}$ at step ‘j’, is $N_{j}=\sum_{k=1}^{k=j}2^{k-1}=2^{j}-1$.

The concentration at each step is computed by using the following relationship between values of the concentrations at successive steps, $C_{j}$ with a set of $N_{j}$ lines and $C_{j+1}$ with $N_{j+1}$ lines. Denote the contribution to the concentration of the $k^{th}$ line source created in the $j^{th}$ step by $f_{j}^{k}.\;$ Then, the contribution, $c_{j}$, of the set of lines, $n_{j}$, created in the $j^{th}$ step to the concentration, $C_{j}$, created in j steps is
(9)$$c_{j}=\frac{Q}{p_{j}}\sum_{k=1}^{k=n_{j}}f_{j}^{k}\;\mathrm{where}\;p_{j}=\sum_{k=1}^{k=n_{j}}l_{j}{}^{k}.$$
where $l_{j}^{k}$ is the length of the $k^{th}$ line source created in step j. The total length of the line sources inserted in the $j^{th}$ step is $p_{j\;}$. Then, the concentration, $C_{j}$, accumulated over j stages follows from Equation (9)
(10)$$C_{j}=\frac{\sum_{k=1}^{k=j}c_{k}p_{k}}{\sum_{k=1}^{k=j}p_{k}}\;\mathrm{or}$$
(11)$$C_{j}L_{j}=\sum_{k=1}^{k=j}c_{k}p_{k}\;\mathrm{where}\;L_{j}=\sum_{k=1}^{k=j}p_{k}$$

Note that sum of the lengths of all the line sources generated in j steps is $L_{j}$. The concentration, $C_{j}$, resulting from j  steps is computed by combining the concentrations, $c_{k}$, weighted by the total length of the lines, $p_{k}$, generated in the $k^{th}$ step. We see from Equation (10) that computing $C_{j+1}$ only requires calculation of the contribution, $c_{j+1}$, to combine with $C_{j}$, computed in the $j^{th}$ step, to yield
(12)$$C_{j+1}=\frac{1}{L_{j+1}}\left(C_{j}L_{j}+c_{j+1}p_{j+1}\right).$$

Computational efficiency is achieved by computing only the contribution from the line sources generated at that stage and combining it with the sum of the contributions from the previous stages (Equation (12)). At each step, the total emission rate from the source is distributed over the sum of the lengths of the line sources that exist at that step. This ensures that the total emission rate, Q, is conserved between steps; the emission rate per unit length of the line source is uniform across the line sources contributing to the concentration, $C_{j}$, at each of the j steps. The updating is terminated when the relative error, $\left|C_{j+1}-C_{j}\right|/C_{j}$, is less than an error limit, which we take as $10^{-5}$.

The convergence of the sum can be sped up by treating the sum as an integral expressed numerically using the trapezoidal rule [17]. After a specified number (5 in our case) of successive integrals are computed, they are used to compute the value of the integral at zero spacing between lines using fifth order polynomial extrapolation based on Neville’s algorithm [17]. If the relative interpolation error is greater than $10^{-5}$, another integral is computed by updating the previous value of the integral. This value is then used with the previous 4 values to estimate the value at zero spacing and the corresponding relative error. We find that seven updates are sufficient to obtain convergence of the integral within the specified error of $10^{-5}$.

## 5. Results

Figure 5 and Figure 6 show the concentration patterns associated with different shaped area sources for two wind directions $180^{{}^{\circ}}$ and $225^{{}^{\circ}}$. The dimensions of the area sources and the surrounding domain are in arbitrary units. The area of all the sources is 24 units, and the emission rate, Q, is 1000 units. The associated micrometeorology are wind speed, U=5 m/s  at a reference height, $z_{ref}=3\;m$, friction velocity, $u_{*}=0.3\;m/s$, standard deviation of horizontal cross-wind velocity fluctuations, $\sigma_{v}=0.6\;m/s$, and Monin–Obukhov length, L=−30 m. The release is at 1 m surface, and the concentrations are also computed at z = 1 m.

The magnitude of the concentrations outside the area source is of the order of $Q/Du_{*}x$, where D is the dimension of the area source perpendicular to the wind vector, and x is the distance from the source. The values of the concentrations outside the contour lines are less than the minimum value shown in the contour labels.

The panels show that the changing the shape of the area source changes the concentration patterns both within and downwind of the area source. Note that all the area sources have the same total emission rates and emission densities. The L-shaped source yields a substantially different concentration pattern than that of the other two area sources shown in Figure 5 and Figure 6.

Figure 6 shows that the downwind concentration patterns associated with the rectangle and polygon shaped area sources are similar when the wind direction is $225^{{}^{\circ}}$. However, as expected the L-shaped area source produces a significantly different concentration pattern with steep gradients close to the source with the concentrations falling off more rapidly because of the larger effective width of the source compared to the other two.

It is clear that there is no straightforward method that would allow us to represent the polygon or the L-shape by an equivalent rectangle to reproduce the concentration pattern associated with these shapes; it is necessary to account for the shape of the area source to model the downwind impact of the source. In the next section, we demonstrate a real-world application of the area source model.

## 6. A Real-World Application

The area source model described in this paper was used to analyze data collected in field studies at a dairy farm located in southern California. The objective of these studies was to estimate methane emissions from five manure lagoon ponds (Figure 7). The liquid manure stream enters the right most pond outlined in red in Figure 6 and flows sequentially through the remaining ponds to the left by gravity.

Measurements of methane mixing ratios were made with an instrumented mobile platform, which stopped at several locations around the manure lagoon (Figure 8) on 14 August 2018, between 10:54 and 15:44. The sampling was conducted for about 9 min at each location for a total of 29 receptors. A 3-D sonic anemometer located on a tower upwind of the manure lagoon provided onsite meteorological information for dispersion modeling. The methane mixing ratios were measured with a cavity ring-down spectrometer (Picarro 2210-i) at a height of 2.87 m.

Initially, the mobile platform circulated around the whole lagoon complex. Preliminary modeling indicated that the red highlighted lagoon in Figure 7 contributed to more than 95% of the total methane emissions, consistent with the expectation that the highest emissions should come from the lagoon with the greatest volume of volatile solids (fresh manure). Subsequent measurements and modeling focused on quantifying emissions from just this lagoon.

The area source dispersion model was used to estimate emissions through the relationship that relates the measured atmospheric methane concentration (mixing ratio) at any receptor ‘*j*’ to the corresponding model estimate as
(13)$$C_{j}=C_{bg}+\sum_{i}E_{i}T_{ij}+\varepsilon_{j}$$
where $T_{ij}$ is the modeled impact of source ‘*i*’ on receptor ‘*j*’ using a unit emission rate, $E_{i}$ is the unknown emission rate from source ‘*i*’, and $\varepsilon_{j}$ is the residual. The background concentration, $C_{bg}$, is also treated as an unknown. The emissions and the background concentrations are the values that minimize $\sum_{j}\varepsilon_{j}^{2}$ with the constraint that their values are greater than or equal to zero. To achieve this, we use the MATLAB function *lsqnonneg* [18].

The emission rates from the manure lagoon are inferred using two configurations of the lagoon: (1) as a homogeneous source and (2) as a heterogeneous source with four areas (Figure 8). Table 1 shows the emission rates inferred by fitting model estimates to the corresponding measurements of methane concentrations. The 95% confidence intervals for these emission rates and background concentration are computed through a version of bootstrapping: the differences between the residuals $\varepsilon_{j}$ and the mean residual $\langle \varepsilon_{j}\rangle$ are added randomly to the best fit model estimates to create 1000 sets of pseudo-observations, which are then fitted to the model estimates to create a distribution of emission rates and background concentrations. This computationally demanding exercise, described in more detail in the paper by the authors of [10], is facilitated by the computationally efficient area source model described in this paper. The emission rates, presented in Table 1, compare well with those estimated with the more commonly used backward Lagrangian particle model (LPM, [19]) [10].

Figure 9 shows the performance of the model in describing the measured methane concentrations in the two configurations; the left panels show how closely the model estimates follow the measurements, and the right panels are scatter plots. When the lagoon is treated as a homogeneous source, the model performs poorly in describing the spatial pattern of measurements, and the $R^{2}\;$ is only 0.14. Model performance improves substantially when the lagoon is treated as four areas with different emission rates. The $R^{2}$ between model estimates and the corresponding measurements is 0.85, and the $m_{g}$, the mean of ratios of the modeled to measured values, is 1.01, indicating little bias in the model.

When the lagoon is treated as a single homogeneous source, the relatively poor description of the measurements provided by the model is reflected in the large 95% confidence interval of the inferred emission rate (Table 1). The smaller confidence interval for the heterogeneous treatment of the lagoon is consistent with the better performance of the model.

The highest inferred emission rate, 204 kg/d, occurs in the top right section of the lagoon (Figure 8 (Right), Table 1). This inference is consistent with the direction of the flow of the manure through the lagoon. The manure entering the lagoon from the top right corner disturbs the surface of the lagoon, which leads to emissions from the deeper layers of the lagoon. The concentrations have their highest values (Figure 8) at this location suggesting the highest emission rates.

The concentration of the receptor at the top left corner of Figure 8 is the background concentration for the model domain. The measured background concentration of 2.2 ppm compares well with the estimated background concentration of 2.4 ppm.

## 7. Summary and Discussion

This paper brings together ideas from past work to propose an approach that can be applied to area sources that can be represented by a polygon. The method can estimate concentrations within and outside the area source with no restrictions on the distance of the receptor from the source. We demonstrate the usefulness of the efficient area source algorithm by applying it to infer emissions rates of methane from a manure lagoon located in a dairy.

The area source is described by the coordinates of the vertices of the polygon used to describe the shape of the source. It is only necessary to specify the total emission rate from the source, unlike in the area source algorithms currently used in models such as AERMOD, which require the emission rate per unit area of the source. The method incorporates numerical methods that make it computationally efficient enough to use bootstrapping to estimate the uncertainty in emission estimates derived from inverse modeling.

The numerical solution of diffusion Equation (4) offers a convenient approach to modeling vertical dispersion of near surface releases. Some advantages it has over the traditional plume based approach are (1) the profile of measured horizontal velocity can be used in the solution if the Monin–Obukhov similarity profile is not believed to hold at a site. (2) It is straightforward to impose boundary conditions (Equation (5)) involving deposition velocity at the surface and zero flux at the top of the boundary layer. (3) It does not restrict the concentration profile to acquire a Gaussian shape, and (4) it is at least as numerically efficient as the Gaussian plume approach. We do realize that the diffusion equation does not provide a realistic description of dispersion of elevated releases.

The application of the model to area sources with different shapes but with the same emission rates and emission densities yields significantly different concentration patterns downwind of the sources. This suggests that there is no straightforward method to reduce different shaped area sources to an equivalent shape to estimate downwind concentrations.

## Figures and Tables

**Figure 1 ijerph-20-06110-f001:**
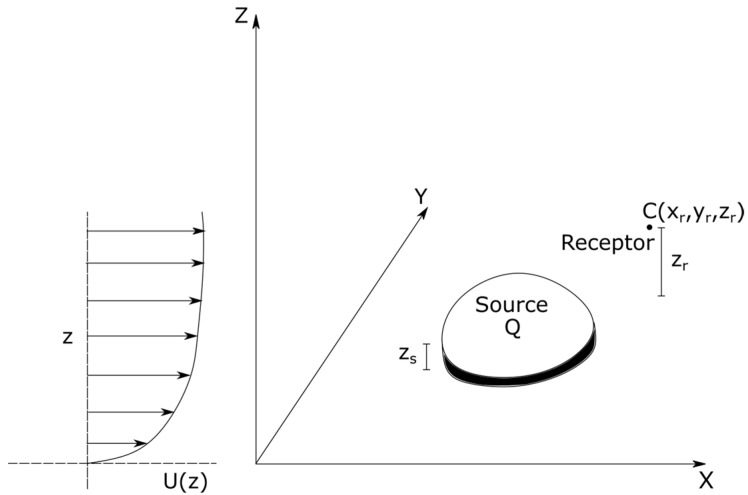
Illustration of the area source being modeled.

**Figure 2 ijerph-20-06110-f002:**
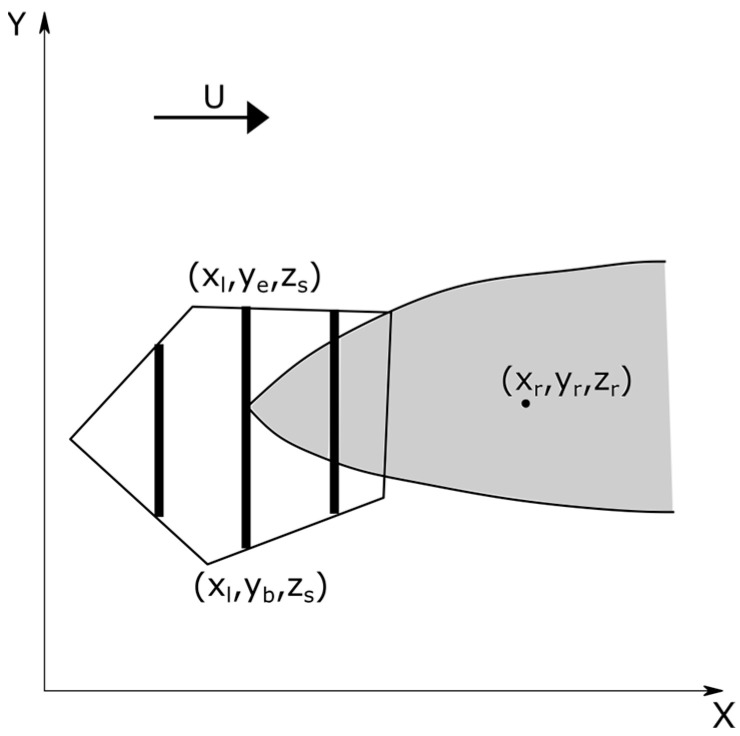
Area source is represented by a set of three equally spaced line source (solid black lines) that are perpendicular to the surface wind. The black arrow represents the near surface wind direction. The *z*-axis is normal to the plane of the figure.

**Figure 3 ijerph-20-06110-f003:**
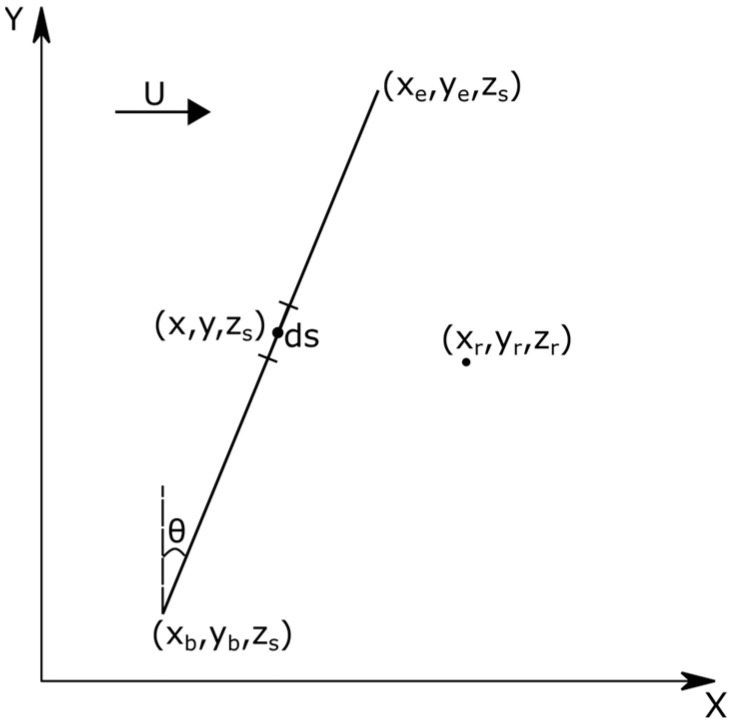
Coordinate system used to calculate the contribution of a point source at $\left(x,y,z_{s}\right)$ to the concentration at $\left(x_{r},y_{r},\;z_{r}\right)$. The system X−Y has the X -axis along the mean wind direction, which is at an angle θ to the line source. The *z*-axis is normal to the plane of the figure.

**Figure 4 ijerph-20-06110-f004:**
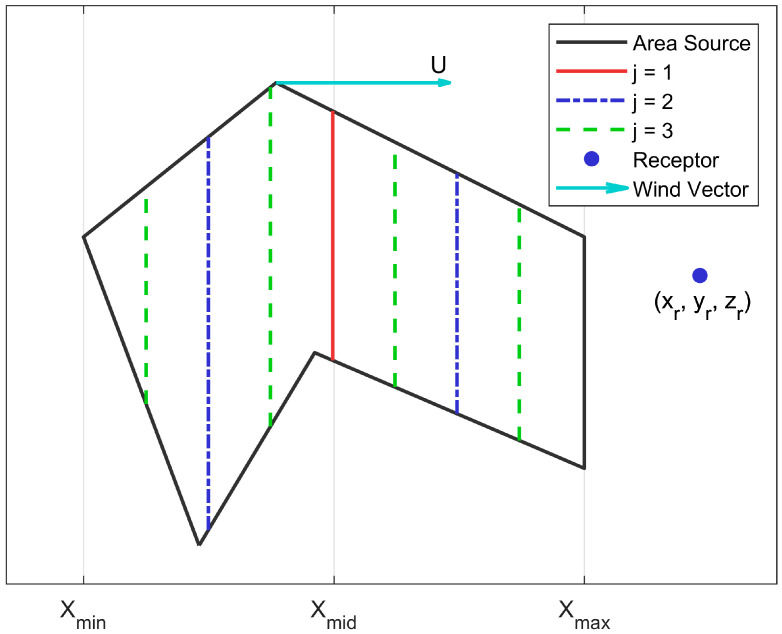
Illustration of the steps involved in evaluating the contributions of the line sources to the concentration at a receptor. At j=1, a single line source is generated (solid red line) at $x_{mid}$. At j=2 two additional line sources are generated (dashed and dotted blue lines) leading to a total of 3 line sources. At j=3, four additional line sources are generated (dashed green lines) leading to a total of 7 line sources.

**Figure 5 ijerph-20-06110-f005:**
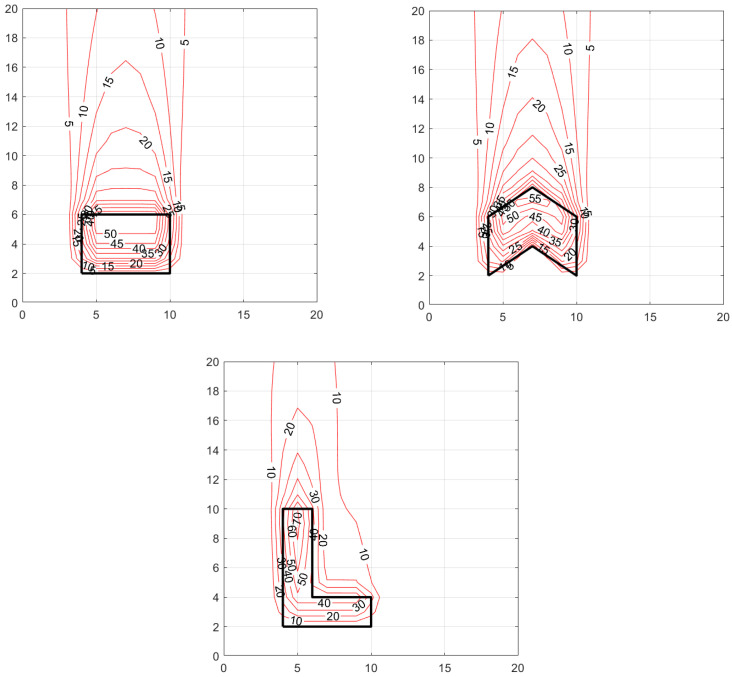
Concentration patterns resulting from an area source with different shapes with a wind direction of $180^{{}^{\circ}}$. Meteorological variables are U=5 m/s, $u_{*}=0.3\;m/s$, $\sigma_{v}=0.6\;m/s$, L=−30 m, $z_{ref}=3\;m$.

**Figure 6 ijerph-20-06110-f006:**
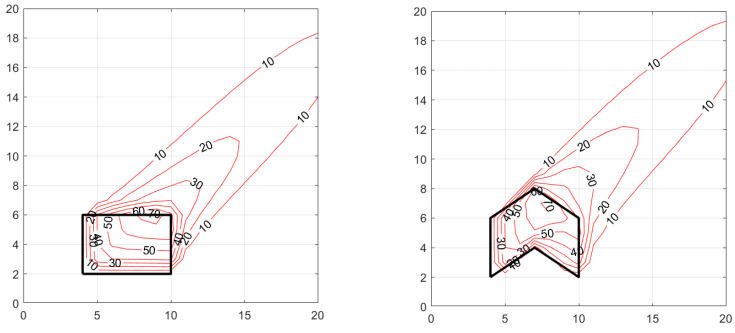
Concentration patterns resulting from area sources with different shapes with a wind direction of $225^{{}^{\circ}}$. Meteorological variables are U=5 m/s, $u_{*}=0.3\;m/s$, $\sigma_{v}=0.6\;m/s$, L=−30 m, $z_{ref}=3\;m$.

**Figure 7 ijerph-20-06110-f007:**
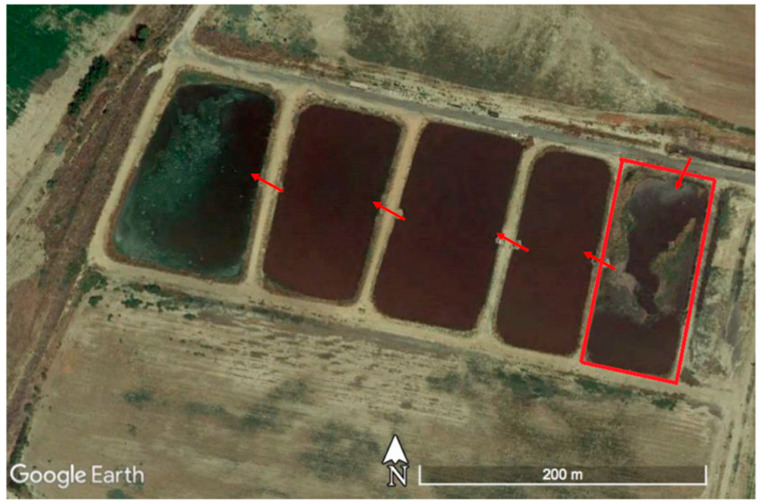
Aerial view of the manure lagoon complex at the southern California dairy. Red arrows show the flow of the manure.

**Figure 8 ijerph-20-06110-f008:**
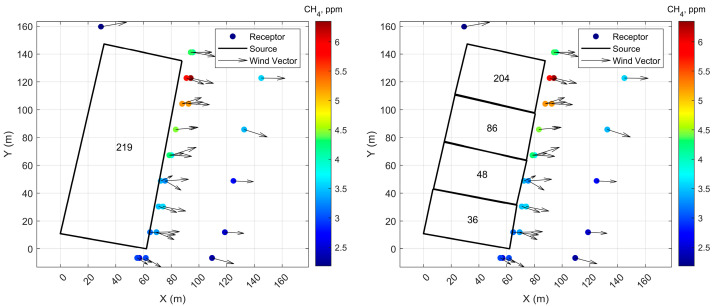
The left panel shows the model configuration considering the lagoon to be a homogenous source, while the right panel shows the model configuration considering the lagoon as an heterogenous source. The circles, color coded by concentration, represent the locations of methane concentration measurements. Black vectors represent average wind direction at each receptor during the measurement period. The number within each box is the estimated methane emission rate in kg/d. See also Table 1.

**Figure 9 ijerph-20-06110-f009:**
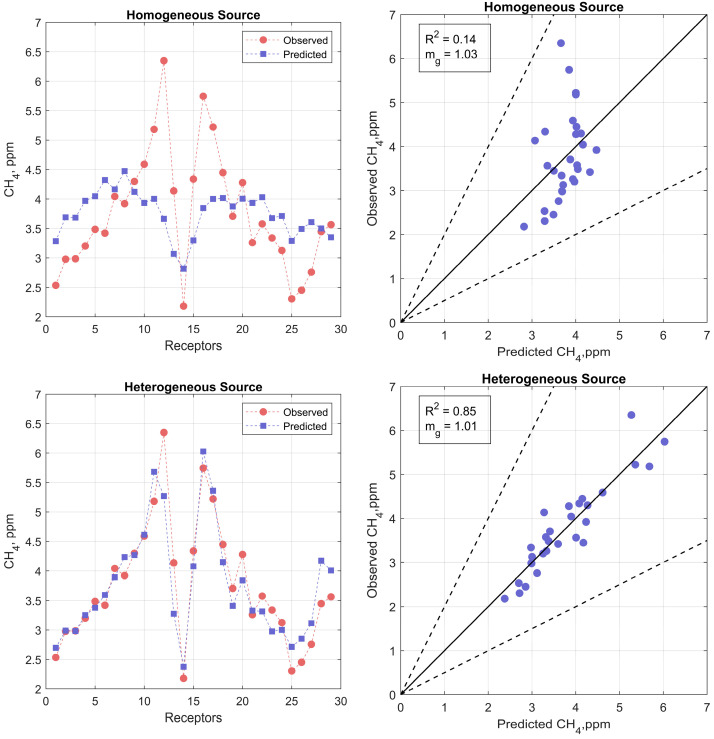
The top panels show the performance of the dispersion model in describing the measurements when the lagoon is treated as a homogeneous source. The bottom panels correspond to the lagoon treated as a heterogeneous source of methane. The left panels show how closely the model estimates follow the measurements. The right panels show scatterplots of model predicted and observed concentrations. The dashed lines around the one-to-one lines in the right panels enclose model estimates within a factor of two of the measurements.

**Table 1 ijerph-20-06110-t001:** Inferred emissions and their 95% confidence intervals (CI) from the lagoon. The values within the brackets are the 95% CI normalized with the best fit value.

Configuration	Source	Emissions (kg/d)	95% Confidence Interval (kg/d)	
			Lower Limit	Upper Limit
Homogenous Source	**1**	**219**	**15 (0.1)**	**404 (1.8)**
	BG (ppm)	2.8	2.0 (0.7)	3.7 (1.3)
Heterogenous Source	1	36	2 (0.0)	71 (1.9)
	2	48	24 (0.5)	72 (1.5)
	3	86	58 (0.7)	120 (1.4)
	4	204	166 (0.8)	242 (1.2)
	**Mean Sum**	**375**	**275 (0.7)**	**470 (1.3)**
	BG (ppm)	2.4	2.0 (0.8)	2.8 (1.2)

The bold numbers are the overall emission rates from the two configurations used.

## Data Availability

Data shared upon request.

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
