# Peer review of "Accounting for Area Sources in Air Pollution Models"

_ijerph, 2023, doi:10.3390/ijerph20126110_

Round 1

Reviewer 1 Report

The paper describes a methodology to model dispersion of airborne pollutants from area sources.

The paper, also according to remarks of the authors, is somewhat a review of existing methods that are well described and checked both in a theoretical set of simulations and in a case study. Although a review, the excursus is linear and sound and would be useful especially for “newbies” or for researchers who would check and compare methodologies.

In the end, the paper is well organized and well written and deserves publication. Given the nature of the paper, I would suggest the following amendments.

Enhance references by adding more up-to-date papers.

Explain better equations including the explanation of all the terms, even well-known ones. For instance in Eq.1, many terms are not described including dispersion parameters while ‘Fz’ is only introduced without a proper description

Figure regarding results and discussion should be enlarged and better explained, especially figs 3-5.

Reviewer 2 Report

This study summarized the core knowledge representing area source, which is already implemented in several dispersion models. The message is concise and worthy to read. However, there are many missing information throughout the manuscript. Comprehensive revision is required for publication in IJERPH.

1.    As the paper does not deal with full aspects of dispersion modeling, the title should be more specific.

2.    [Abstract, Introduction] Highways are not normally represented as area sources.

3.    [Introduction] The first three sentences are exactly duplicated. Delete them.

4.    The introduction section doesn’t deliver comprehensive research background and purpose of this study. There should be the literature survey that can give an important information on the representation of area emission source.

5.    Figure 1 does not have a z-axis, while the points are 3-D representation(x,y,z).

6.    It is very confusing that Equation 1 have z-axis again. Dimension should be clearly defined.

7.    It is hard to understand that all individual line sources have the same wind speed of U(z) even at different locations? Then, it seems identical to the existing representation of area source.

8.    Equation 8 is actually two separate equations. Divide it.

9.    Explanation and discussion on Figure 4 and 5 are very poor. Give some interpretation of the results compared to Figure 3.

10. The first two paragraphs of section 3.1 are proper to be moved to the Introduction section.

11. Provide the necessary information on the site and measurement specification of the manure lagoon complex. No information is given on date and location.

12. In table 1, where did you measure the BG concentration of methane? If so, indicate it in Figure 6. Otherwise, the estimated BG concentration should be verified.

13. Provide No. of receptors in Figure 6 or 7.

14. How did you know the atmospheric stability of lagoon site? Provide the information.

Reviewer 3 Report

The paper “Modeling Dispersion from Area Sources” focuses on a current issue.  The approach proposed in the paper has been incorporated into the version of the AERMOD and the Danish OML models and can be used in the suggested version for air pollution modelling.

Here are some comments that should be considered before publishing:

ü  The title of the paper is very general. It is recommended to change the title to take into account the issues of modeling air pollution.

ü  Lines 23-24- “[1] modeled an urban area as set of 23 area sources, each of which is represented as a grid square” – such reference to the article is not correct.

ü  Lines 32-33 – “A few papers [2,3] have presented more accurate approaches to modeling area 32 sources” – it should be mention a little more about these models.

ü  It is recommended to extend the bibliography with more scientific publications in the field of the discussed topic.

ü  Out of a rather modest bibliography, in which 18 scientific publications were quoted, as many as 6 belong to the authors of the presented manuscript and in this form the manuscript should not be published.

Reviewer 4 Report

The manuscript looks like a technical report or a chapter of the book. Furthermore, it does not provide a new contribution/originality to the literature.

“Several approaches have been used to model dispersion of emissions from such sources. [1] modeled an urban area as set of area sources, each of which is represented as a grid square.” “We suggest the solution of the two-dimensional diffusion equation as an alternative because it offers several advantages that are discussed in this section. [8] show that the solution provides an excellent description of the vertical distribution of tracer concentrations measured during the classic Prairie Grass experiment when the wind speed and eddy diffusivity are formulated using Monin-Obukhov similarity; the solution, unlike the Gaussian distribution, describes the evolution of the distribution from nearly Gaussian during stable conditions to nearly exponential during unstable conditions. “Dot in the sentence should be removed.

There is no information on the determination of methane emissions from the area sources in the introduction part. The introduction part should be improved and the previous studies should be compared in terms of shortcomings.

Round 2

Reviewer 3 Report

Accept in present form

Reviewer 4 Report

The revised Manuscript has been improved and it is acceptable for publication.